# Soft-Tissue Healing Assessment after Extraction and Socket Preservation Using Platelet-Rich Fibrin (PRF) in Smokers: A Single-Blinded, Randomized, Controlled Clinical Trial

**DOI:** 10.3390/diagnostics12102403

**Published:** 2022-10-03

**Authors:** Yasser Alrayyes, Saleh Aloraini, Ahmed Alkhalaf, Reham Aljasser

**Affiliations:** 1Dental University Hospital, King Saud University Medical City, Riyadh 11545, Saudi Arabia; 2Department of Periodontics and Community Dentistry, College of Dentistry, King Saud University, Riyadh 11545, Saudi Arabia; 3Prince Sultan Military Medical City, Riyadh 12233, Saudi Arabia

**Keywords:** platelet-rich fibrin, regeneration, alveolar-ridge preservation, wound healing

## Abstract

Background: Wound healing is essential in any surgical procedure, and multiple factors, such as smoking, can impair it. The aim of this clinical trial was to evaluate the ability of platelet-rich fibrin to enhance socket wound healing in smokers. Methods: A total of 18 smoker participants with forty non-restorable upper molars indicated for extraction were recruited to the study and were randomly allocated to four different groups for the assessment of four techniques: advanced platelet-rich fibrin (A-PRF), factor-enriched bone graft matrix with advanced platelet-rich fibrin (A/S-PRF), freeze-dried bone allograft and crosslinked collagen membrane (FDBA/CM), and resorbable collagen plug (RCP). One examiner clinically measured soft-tissue closure and the healing pattern using a periodontal probe and a healing index. Each subject was given a questionnaire after each follow-up visit to record several patient-reported experience measures (PREMs). This was conducted at baseline and 10, 21, and 28 days after the extraction procedure. Results: Both A-PRF and A/S-PRF showed significant results in terms of mesio-distally (*p* = 0.012), and healing pattern parameters (*p* < 0.0001), while RCP showed the least favorable outcome. Conclusions: Different forms of PRF exhibited enhanced wound closure and healing patterns, as well as reduced post-operative complications among smokers.

## 1. Introduction

Wound healing is an essential aspect after dental extraction or any surgical procedure; it is defined as the complex and dynamic process of replacing devitalized and missing cellular structures and tissue layers [1]. Four essential phases constitute normal socket wound healing, i.e., hemostasis, inflammation, proliferation, and resolution [2]. Several factors may influence these four phases, such as general health conditions, including uncontrolled diabetes [3]; radiation; chemotherapy; antiresorptive medications [4]; and smoking [5]. Any disturbance to wound healing may result in excessive bleeding or in the absence of blood-clot formation, as seen in alveolar osteitis [6].

The dermis and the oral mucosa exhibit similar structural features, comprising an epithelial layer supported by underlying supportive connective tissue [7]. Although both structures are relatively similar, some subtle differences are still present, such as the environment they are exposed to; the oral mucosa is constantly exposed to saliva, which acts as a protective barrier that controls the microbiota in the oral cavity [8]. Another aspect of saliva is its ability to enhance wound healing, as it contains several growth factors, such as vascular endothelial growth factor (VEGF) and fibroblast growth factors (FGFs), which induce faster re-epithelialization, angiogenesis, and maturation [9].

The effects of smoking on wound healing following many surgical interventions have been extensively evaluated. It has been shown that smoking can negatively affect the healing process after flap surgery and impair regeneration outcomes [10]. These negative impacts have been attributed to arteriolar vasoconstriction and decreased blood flow, the mutation of gingival fibroblasts, and cell differentiation and migration caused by smoking. Heat and toxic products, such as nicotine, carbon monoxide, and hydrogen cyanide, are the primary causative factors of such impairment [5]. It is also crucial to note that smoking modulates the pro-inflammatory and anti-inflammatory markers that play a role in periodontal-tissue wound healing [11,12,13,14]. Smoking can affect the immune system in various ways, which leads to immunosuppression and increased susceptibility to infection [15]. Long-term exposure to smoking and nicotine intake can suppress B-lymphocyte development, proliferation, and immune functions [16]. These changes can affect the levels of immunoglobulins in both serum and saliva [17]. 

Several attempts have been made to facilitate post-extraction healing, including using platelet-rich fibrin (PRF), the most studied type of platelet concentrates for accelerating wound healing, especially for soft-tissue migration. PRF can reduce the inflammation of the periodontal tissues, preserve the alveolar site, and improve alveolar-bone defect repair, thereby enhancing bone regeneration [18]. Although some studies have reported no significant differences when using PRF, most have reported improved soft- and hard-tissue regeneration when using PRF alone or with other grafting material [19].

Numerous clinical trials have been conducted to assess PRF potential in socket preservation procedures; it has been proved that PRF alone or mixed with freeze-dried bone allograft (FDBA) could minimize bone remodeling post-extraction and enhance new bone formation [20,21]. Several systematic reviews have also concluded positive results for PRF when used alone or with other grafting materials on soft-tissue healing and bone regeneration [22,23]. However, limited data are available on the effect of PRF on soft-tissue healing among smokers. 

Therefore, the present study aimed to evaluate the PRF effect on soft-tissue healing and compare healing patterns in extraction sockets following two different PRF applications with conventional grafting, using a freeze-dried bone allograft (FDBA) and a crosslinked collagen membrane or using a resorbable collagen plug, only in smokers after dental extraction and alveolar-ridge preservation (ARP) procedures. 

## 2. Materials and Methods

### 2.1. Ethical Statement

The study protocol was approved by the institutional review boards (IRBs) of King Saud University Medical City (approval No. E-21-5999) and College of Dentistry Research Centre (CDRC) (approval No. PR 0122), and patients’ informed consent was obtained. In addition, the study was registered at clinicaltrials.gov with ID NCT05161455 and was conducted in accordance with the requirements of the Helsinki Declaration of 1975 (revised in October 2013). All individuals provided informed written consent to participate in the trial and were informed of their ability to withdraw from the study at any given time upon their preference.

### 2.2. Participant Recruitment

Patients indicated for the extraction and ridge preservation of upper molar teeth for future single implant placement were enrolled at the Department of Periodontology at Dental University Hospital, King Saud University, Riyadh, Saudi Arabia, between June 2021 and April 2022.

### 2.3. Inclusion Criteria and Exclusion Criteria

Participants were recruited according to the following criteria: (1) they were current heavy smokers (defined as 10 or more cigarettes/day) [24]; (2) they were 21 years of age or older; (3) they had an upper molar indicated for extraction due to extensive decay, tooth fracture, or failed root canal treatment; and (4) their extraction socket maintained a four-wall configuration.

Participants were excluded when (1) they were unable to undergo oral surgery procedures; (2) they had an active periodontal disease; (3) they had teeth with endodontic peri-apical lesions; (4) they had an active dental infection in the bone; (5) they had ankylosed teeth; (6) they were women who were pregnant or nursing a child; (7) they had compromised health issues that could have affected the ability of the patient’s tissues to heal, such as uncontrolled diabetes; (8) they had medical conditions affecting normal bone formation (i.e., osteoporosis) or took medication (i.e., bisphosphonate) associated with compromised bone healing; and (9) they had inadequate oral hygiene.

### 2.4. Sample Size Calculation

The sample size was determined using G Power software, where the confidence level was set at 95% and the power level at 0.5 with a moderate effect size. The final targeted sample size was thirty socket sites (n = 30), and the number was rounded up to forty (n = 40) to accommodate an attrition rate of 10% and to have equal distribution among groups. 

### 2.5. Participant Allocation, Randomization, and Blinding

Every extracted socket site was randomly allocated to one of the following four groups:Group I (A-PRF)—advanced platelet-rich fibrin;Group II (A/S-PRF)—factor-enriched bone graft matrix (commonly known as sticky bone) using autologous fibrin glue combined with freeze-dried bone allograft (FDBA), cortico-cancellous blend (250–1000 µm) (Citagenix, Montreal, QC, Canada), and an advanced platelet-rich fibrin membrane (A-PRF) to cover it;Group III (FDBA/CM)—freeze-dried bone allograft (FDBA), cortico-cancellous blend (250–1000 µm) (Citagenix, Montreal, QBC, Canada), and a crosslinked collagen membrane (Citagenix, Montreal, QC, Canada) serving as a positive control;Group IV (RCP)—resorbable collagen plug (Citagenix, Montreal, QC, Canada) alone; Group IV served as a negative control group. Each site was randomly assigned to one of the four groups that followed different ridge preservation approaches. A simple randomization method using sequence generation was applied by one periodontist (R.J.). All participants were blinded to which grafting method was conducted.

### 2.6. Initial Visit

At the screening visit, the participants were asked to read the study information and sign their informed consent and were encouraged to ask any questions related to the study. Via periodontal examination, data regarding periodontal pocket depth (PPD), bleeding on probing (BOP), and plaque index (PI) were recorded for the tooth intended for extraction and the two adjacent teeth. Periodontal probing depth was measured by inserting a University of Michigan O Probe with Williams marking (Hu-Friedy^®^ Mfg, Chicago, IL, USA) within the sulcus area with gentle pressure (0.75 Ncm) at three buccal points and three palatal points. Bleeding on probing (BOP) was assessed by utilizing the Ainamo and Bay index. The assessment was based on either the presence (+) or absence (−) of bleeding at the probing site immediately after measuring the periodontal pocket depth. The index number was calculated by dividing the number of bleeding points by the total number of present teeth multiplied by 100 [25]. As for plaque evaluation, the O’Leary index was utilized by asking each patient to rinse using a disclosing solution (Hu-Friedy Mfg. Co., Chicago, IL, USA) and then examining each tooth surface, except for the occlusal surfaces; finally, we divided the number of stained surfaces by the total number of surfaces scored and multiplied by 100 [26].

In addition, oral hygiene instructions were provided, and scaling and root planning was performed two weeks before surgery [27]. All non-surgical treatment was conducted by the same periodontist (Y.R.).

### 2.7. PRF Preparation

An updated protocol was followed to prepare A-PRF and sticky bone [28]. For A-PRF preparation, a blood sample of 10 mL per tube was collected from the patient with no separating gel nor anticoagulant (two tubes were obtained and were red-coded). The blood withdrawal time was 15 s per tube and was immediately centrifuged, according to the following protocol: 13 × 100 rpm for 14 min.

After centrifugation, there were three components in the vial: red blood cells at the bottom; a fibrin clot, representing PRF, in the middle; and acellular plasma at the top. PRF was obtained by extracting the matrix from the vial with tweezers and removing the red clot.

Furthermore, the compression of the PRF piece to form a flat membrane through a slow and homogeneous gentle squeeze was conducted, and the final membrane was maintained homogeneously wet and soaked in serum using L-PRF Wound Box.

L-PRF Wound Box^®^ (PRF box-mk2; Avtec surgical, Mt Pleasant, SC, USA) was composed of a metal container of 17.5 cm × 7.6 cm 2 cm containing a perforated steel plate of 150 m × 68 m × 1.5 m. A second steel plate acted as a compressor; it had dimensions of 150 mm × 68 mm × 1.5 mm and weighed 148 g. This plate exerted a pressure of 142.437 Pa/cm^2^. Both plates were used to prepare the PRF membrane in this study.

As for sticky bone preparation, one S-PRF tube (green-coded) was used for blood withdrawal. The collected blood was centrifuged in the same manner, and S-PRF liquid was then taken directly from the tube using a 2 mL syringe; then, it was mixed with FDBA particles. All blood withdrawal and PRF preparations were made by a trained and calibrated dental nurse from the Department of Periodontology [28].

### 2.8. Surgical Protocol

Each patient was given a prophylactic dose of two grams of amoxicillin (SPIMACO, Al-Qassim, Saudi Arabia) according to the 2017 American Dental Association guidelines one hour before surgery. In case of an allergy, 500 mg of azithromycin (Riyadh Pharma, Riyadh, Saudi Arabia) was given instead [29]. Local anesthesia with 2% Xylocaine injection with 1:100,000 epinephrine was then given as buccal and lingual infiltration (DENTSPLY Pharmaceutical, York, PA, USA). A sulcular incision was performed using a 15c surgical blade (Glassvan Surgical Blade, Niraj Industries, Mumbai, India), and atraumatic extraction was initiated using a periotome (Hu-Friedy^®^ Mfg, Chicago, IL, USA) to separate the supra-crestal periodontal ligament fibers with no flap reflection. The tooth was luxated using straight elevators and extracted with forceps (Hu-Friedy^®^ Mfg, Chicago, IL, USA) to minimize stress on the buccal bone in the most atraumatic manner possible. The extraction socket was then debrided, and granulation tissue was removed using a Lucas surgical curette (Hu-Friedy^®^ Mfg, Chicago, IL, USA); then, the socket was irrigated with saline [30]. In the A-PRF group, the socket was filled with a PRF membrane carved to fit the extraction site and placed just under the gingival margin. In the A/S-PRF group, the socket was filled with a combination of autologous fibrin glue obtained from the green tube and FDBA particles in a 1:1 ratio (sticky bone); then, it was covered with a PRF membrane. In the FDBA/CM group, the socket was filled with FDBA particles and covered with a crosslinked collagen membrane (Citagenix, Montreal, QC, Canada), and in the RCP group, the socket was filled with a resorbable collagen plug (RCP) (Citagenix, Montreal, QC, Canada). All the grafted materials were secured in place using a hidden X suture to minimize the reduction in the width of keratinized tissue [31] using 4.0 silk. All the surgeries were performed by one periodontist (Y.R.).

### 2.9. Post-Operative Instructions

All the participants were instructed to rinse twice daily with chlorhexidine mouth rinse at 0.2% (Avohex; Avalon Pharma, Riyadh, Saudi Arabia) from the day after surgery for 10 days and to avoid brushing or flossing at the surgical site until the day of the second follow-up; furthermore, each subject was prescribed 400 mg ibuprofen to be taken if needed. Finally, all the participants who proved not to be allergic to penicillin were prescribed 500 mg of amoxicillin to be taken three times per day for 7 days.

### 2.10. Assessment of Soft-Tissue Closure

The assessment of soft-tissue closure at the extraction sites was conducted at four-time points, i.e., immediately after extraction and after 10, 21, and 28 days. The process was conducted using a University of Michigan O Probe with Williams marking (Hu-Friedy^®^ Mfg, Chicago, IL, USA). Moreover, an acrylic stent was fabricated before extraction. Four notches were made to represent the midpoints of the buccal, palatal, mesial, and distal dimensions of the tooth and served as reference points for socket dimension measurements. Furthermore, measurements were conducted by placing the probe exactly perpendicularly and at the mid-portion of the socket both bucco-palatally and mesio-distally on the stent and comparing changes in the inner part of the soft-tissue edges to the nearest 1 mm [32].

### 2.11. Intra-Examiner Reliability

All the clinical assessments and measurements were conducted by a single examiner (Y.R.). Clinical measurements were performed on six randomly selected participants. These measurements were repeated after 10 days; Cohen’s Kappa Score was used to measure the reliability level, and the score was 0.94.

### 2.12. Healing Process

The Landry wound healing index was utilized to assess the tissue healing process at the surgical sites [33]. The assessment was conducted by evaluating specific parameters, including tissue color, bleeding response to palpation, the presence of granulation tissue, the characteristics of the incision margins, and the presence of suppuration at 0, 10, 21, and 28 days. The index classifies the healing pattern based on the color of the soft tissue (pink or red), the presence or absence of suppuration, the amount of bleeding, the presence of granulation tissue, and the exposure of connective tissue into five different categories: (1) very poor, (2) poor, (3) good, (4) very good, and (5) excellent (Figure 1).

### 2.13. Post-Operative Pain Score and Patient-Reported Experience Measures (PREMs)

A visual analog scale (VAS) was used to describe pain [34]. The scores were based on self-reported measures of symptoms recorded with a handwritten mark. It was made up of a line representing a range between the two ends of a scale “no pain” on the left end and the “worst pain” on the right end of the scale.

In addition, patient surgery experience outcomes were evaluated. All participants were asked to fill out a questionnaire with ten different questions with ‘‘Yes or No’’ and “how long” answers. Questions were chosen from two pre-validated surveys [35,36]. After gaining official permission to use both surveys, the questions were translated and re-validated. The participants were asked if surgery had affected their ability to eat, speak, sleep, and their daily activities and if it had caused swelling, a foul smell or taste, bleeding in the area, or any other unmentioned problems, as well as whether or not the procedure had been properly explained by the medical team.

### 2.14. Statistical Analysis

Data were analyzed using SPSS statistical software (version 26.0; IBM Inc., Chicago, OH, USA). Descriptive statistics (means, standard deviation, frequencies, and percentages) were used to describe the quantitative and categorical variables. Non-parametric tests—the Kruskal–Wallis test followed by Dunn’s post hoc test—were used to compare the mean ranks of the B-P and M-D socket dimensions, pain assessment, and healing index values among the 4 study groups (A-PRF, A/S-PRF, FDBA/CM, and RCP) at each of the 4 time points (day 0, day 10, day 21, and day 28). In addition, the Friedman test was used to compare the mean ranks of the B-P and M-D socket dimensions, pain assessment, and healing index values of repeated measures on day 0, day 10, day 21, and day 28 for each of the 4 study groups. A *p*-value of <0.05 was used to report the statistical significance of the results.

## 3. Results

A total of 18 patients with 40 extracted teeth completed the study by attending all follow-ups. The mean age of the participants was 38 years old; they were all male, and none were allergic to amoxicillin. Table 1 shows the mean age of the study participants and the mean number of cigarettes smoked across the four study groups; the mean ages of the two groups (A/S-PRF and FDBA/CM) were higher than those of the other two groups. Out of the mean values of the B-P socket dimensions at baseline and on day 10, day 21, and day 28, the mean values were the highest at baseline; similarly, the mean values of the M-D socket dimensions at baseline were higher than those recorded on the other three days of observation. In addition, the mean values of pain assessment on day 10, day 21, and day 28 are given in Table 1, and the healing index, which was recorded using a five-point scale (very poor, poor, good, very good, and excellent) on day 10, day 21, and day 28 indicated that a higher number of participants showed better healing at all the three time points of observation in the PRF study group (Table 1).

Regarding the bucco-palatal (B-P) socket dimension assessment, the comparison of the mean ranks of the socket dimensions among the four study groups at baseline and on day 10, day 21, and day 28 showed statistically significant differences on day 10 and day 21 (*p =* 0.018 and *p =* 0.016, respectively). The post hoc test showed that the mean rank values of the RCP and FDBA/CM groups were statistically significantly higher than those of the PRF groups both on day 10 and day 21. On the other hand, when both the PRF groups were compared, no statistically significant differences were found between them in none of the measurement time frames. Furthermore, when the mesio-distal (M-D) socket dimensions were assessed, the comparison of the mean ranks of the socket dimensions among the four study groups at baseline and on day 10, day 21, and day 28 also showed statistically significant differences on day 21 and day 28 (*p =* 0.017 and *p =* 0.032, respectively). The post hoc test showed that the mean rank values of the RCP group were statistically significantly higher than those of both PRF groups on day 21. Finally, the values of the A-PRF group were shown to be statistically significantly lower than those of all other groups on day 28 (*p =* 0.012) (Table 2).

The comparison of the mean ranks of pain assessment among the four study groups on day 10, day 21, and day 28 showed statistically significant differences on day 10 and day 21 (*p* < 0.0001 and *p* = 0.001, respectively). The post hoc test showed that the mean rank values of the RCP and FDBA/CM groups were statistically significantly higher than those of the A-PRF and A/S-PRF groups on day 10, while there were no differences between the PRF groups. On day 21, the mean rank of the RCP group was significantly higher than those of the other three groups. Moreover, the comparison of the mean ranks of pain assessment within each of the four study groups across the three time points of observation (day 10, day 21, and day 28) showed high statistically significant differences for two groups (FDBA/CM and RCP; *p <* 0.0001 and *p =* 0.002, respectively). The post hoc test indicated that the mean ranks on day 10 were significantly higher than those at the other two time points and that there were no differences in the mean ranks between the time points of day 21 and day 28 in each of the FDBA/CM and RCP groups.

With respect to the healing index, the comparison of the mean ranks of the healing index among the four study groups on day 10, day 21, and day 28 showed statistically significant differences on day 10, day 21, and day 28 (*p* < 0.0001; *p* = 0.001). The post hoc test showed that the mean rank values of both PRF groups were statistically significantly higher than that of the RCP group on day 10, day 21, and day 28, that is, the healing index was statistically significantly higher in the A-PRF and A/S-PRF groups at all the three time points when compared with the RCP and FDBA/CM groups (*p* < 0.0001). In addition, there were no differences between the FDBA/CM and RCP groups at none of the three time points (day 10, day 21, and day 28) (Table 3).

The distribution of the responses to the 10-item patient-reported experience measures (PREMs) at the four time points (baseline, day 10, day 21, and day 28) showed mostly negative responses for most of the items except for the following question: “Did you need any analgesics?”; all participants in both PRF groups reported that they only needed analgesics for the day of the procedure in comparison with other study groups. In fact, the use of analgesics on the day of surgery in the FDBA/CM group included an average of two pills of 400 mg Ibuprofen, and this was extended to day 28 in the RCP group, with an average of five pills per day. Furthermore, to the question “Did you suffer from swelling in the area”, five participants responded positively in the RCP group, and to the item “Did you notice any foul smell or taste”, only one subject in the FDBA/CM group and two participants in the RCP group responded positively, while no positive responses were reported by the PRF groups. To the item “Did you suffer from bleeding in the area”, all study participants (n = 12) in both PRF groups responded positively, and only four participants responded positively in the RCP group on day 28. In addition, all participants at all four time points of observation responded positively to the item “Did the medical team explain procedure to you”. Furthermore, all the participants that received A-PRF and A/S-PRF did not report any pain or discomfort. Bleeding that lasted for almost an hour was reported by the FDBA/CM group. Six participants in the FDBA/CM group reported moderate pain for a few days that required analgesics, with one participant reporting that the pain persisted for more than ten days. Another complained of a sharp bony edge that cleared after 28 days. In the RCP group, nine participants reported severe pain, of which four reported pain persisting for more than 10 days and one for more than 21 days. One subject reported abscess formation after 3 days post-operatively.

## 4. Discussion

The main objective of the present randomized clinical trial was to compare the effects of two different treatment modalities utilizing two proposed protocols of platelet-rich fibrin (A-PRF and A/S-PRF) and to compare them with more conventional protocols used for alveolar-ridge preservation (FDBA/CM and RCP) by evaluating the pattern of epithelial migration and soft-tissue healing in smoker participants indicated for this procedure. In terms of epithelial migration and soft-tissue closure, the A-PRF and A/S-PRF groups revealed a significantly faster rate of soft-tissue closure and un-eventful healing patterns when compared with the two other groups, with the RCP group presented the least favorable outcome among all groups. In addition, the PRF groups underwent a healing process with less pain and fewer soft-tissue closure complications, including swelling and pus formation. Overall, it is worth mentioning that all the groups in the present study achieved complete soft-tissue closure after 28 days. Both PRF groups achieved this at a faster rate.

Overall, smoking has been studied and proven to have an unfortunate effect on biological tissues [5]. Nicotine, which is a key component in cigarettes, has been associated with negative biological impacts on soft-tissue healing, as it causes hypoxia due to its vasoconstrictive properties [37], and oxygen is needed for cell migration in the area, collagen synthesis, and the bactericidal action of neutrophils [5]. Another component that may cause hypoxia is carbon monoxide; it can bind to hemoglobin faster than oxygen, thus reducing its perfusion to the bloodstream, which causes cellular hypoxia [38].

Smoking has been associated with multiple post-operative complications after dental extractions; a frequently reported complication in smokers is post-operative pain; one clinical study, in particular, reported a significant difference when they compared smokers with non-smokers; however, they reported no differences relatively to the number of cigarettes smoked [39]. Another commonly reported complication is alveolar osteitis. According to a recent systematic review [40], this complication is highly common in smokers who undergo both traumatic and atraumatic dental extractions, with a prevalence rate of 13.2% compared with non-smokers (3.8%). In addition to the above-mentioned complications, smoking has been linked to many other consequences, including trismus, swelling, and infection [39]. Therefore, present ongoing efforts are aimed to find treatments or applications that can overcome the occurrence of smoking-related biological deterioration.

Multiple in vitro studies have explored the effect of PRF on soft-tissue healing; in one study, PRF induced higher fibroblast proliferation than fibrin sealant and recombinant platelet-derived growth factor (PDGF), and it was able to prevent the proteolytic degradation of endogenous fibro-genic factors important for wound healing [41]. In another study, where PRF was applied on fibroblasts, they found that growth factors from PRF were able to induce cell viability and proliferation differentiation [42].

PRF’s potential in soft-tissue healing after dental extraction in compromised patients has been assessed multiple times in the literature. One study evaluated the effect of two different variants of PRF as hemostatic agents in patients with anti-platelet medications; compared with the other test groups, both PRF variants were able to minimize post-surgical bleeding and significantly promote wound healing [43]. Another study evaluated the effect of PRF on osteoporotic patients under oral bisphosphonate treatment. They evaluated the effectiveness of PRF on wound closure, and they found a significant difference in soft-tissue closure when compared with leaving the socket with no treatment [44].

The current study findings were in agreement with those of multiple studies with respect to soft-tissue healing. In a study by Azangookhiavi and his team [45], where they compared PRF with FDBA, they noticed accelerated soft-tissue healing in the PRF group. Similar results were reported by Ahmed and his team [46]; when they compared three different treatment modalities, PRF showed superior results in terms of soft-tissue healing. In multiple systematic reviews [22,23,47] where the potential of PRF was addressed, the positive effect of PRF on accelerating soft-tissue healing was highlighted.

The current study presented several strengths. First, all the participants were male, which allowed us to rule out any gender-based variables, and second, to our knowledge, this was the only known study that only targeted smokers and compared PRF with commonly used grafting materials. On the other hand, multiple limitations were also present in the current study; the most critical one was the lack of a real negative control group. Ideally, the sockets of the patients in the test groups should have been compared to an empty socket without any grafting material to have a better understanding of the investigated effects, but the retarded healing of smokers presented ethical challenges to leaving the socket empty; second, although the sample size was larger than that of most similar studies reported in the literature, it was still considered small.

## 5. Conclusions

According to the findings of this randomized clinical trial, PRF showed superior results in terms of soft-tissue closure, healing, and reducing post-operative pain. Further standardized clinical trials with a larger sample size need to be conducted to confirm these findings.

## Figures and Tables

**Figure 1 diagnostics-12-02403-f001:**
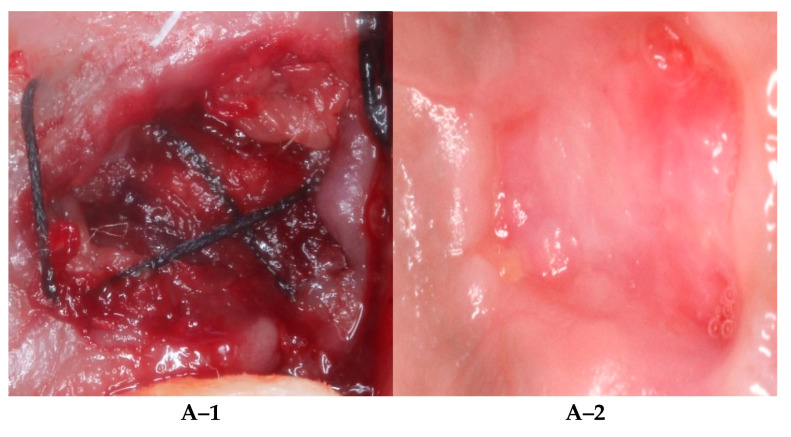
Clinical photos for surgical procedure. (**A–1**) Extraction with A-PRF at baseline (**A–2**) and extraction with A-PRF at 28-day follow-up. (**B–1**) Extraction with A/S-PRF at baseline (**B–2**) and extraction with A/S-PRF at 28-day follow-up. (**C–1**) Extraction with FDBA/CM at baseline (**C–2**) and extraction with FDBA/CM at 28-day follow-up. (**D–1**) Extraction with RCP at baseline (**D–2**) and extraction with RCP at 28-day follow-up.

**Table 1 diagnostics-12-02403-t001:** Distribution of demographic and clinical characteristics of 4 study groups.

Characteristic	Study Groups			
	PRF (A-PRF)	PRF + Sticky Bone (A/S-PRF)	FDBA + Collagen Membrane (FDBA/CM)	Resorbable Collagen Sponge (RCP)
Age (in years)—mean (sd)	34.33 (11.7)	41.58 (11.7)	44.30 (8.2)	32.0 (3.9)
No. of cigarettes—mean (sd)	20.2 (25.4)	22.2 (25.1)	23.0 (27.6)	22.7 (29.1)
B-P socket dimensions (in mm)—mean (sd) At baseline On day 10 On day 21 On day 28	10.9 (2.2)	10.9 (1.7)	10.9 (2.3)	10.9 (2.3)
	4.3 (2.8)	5.3 (2.3)	6.6 (1.3)	7.1 (1.7)
	2.7 (1.9)	3.6 (2.7)	4.6 (2.6)	5.7 (1.6)
	2.2 (2.0)	3.1 (2.3)	4.3 (2.6)	3.9 (1.5)
M-D socket dimensions (in mm)—mean (sd) At baseline On day 10 On day 21 On day 28	8.5 (2.8)	9.6 (1.7)	9.5 (1.6)	9.1 (1.8)
	4.7 (2.9)	6.2 (1.7)	6.8 (2.0)	7.4 (1.7)
	3.5 (2.5)	3.7 (2.1)	5.3 (3.5)	6.7 (1.6)
	2.4 (2.1)	3.1 (2.1)	4.6 (3.2)	5.3 (1.8)
Pain assessment (1–10 scale)—mean (sd) On day 10 On day 21 On day 28	0	0	3.3 (2.1)	3.2 (2.4)
	0	0	0.2 (0.6)	1.3 (1.5)
	0	0	0	0.2 (0.6)

**Table 2 diagnostics-12-02403-t002:** Comparison of mean ranks of B-P and M-D socket dimensions on day 0, day 10, day 21, and day 28 among the 4 study groups.

Time Point	Mean Ranks of B-P Socket Dimensions				Post Hoc Test Statistic	*p*-Value
	A-PRF	A/S-PRF	FDBA/CM	RCP		
Day 0	23.25	22.42	21.05	20.83	0.28	0.964
Day 10	15.29 *	16.42 *	27.30 *	29.83 *	10.05	0.018
Day 21	14.96 *	14.00 *	24.80 *	30.94 *	9.29	0.016
Day 28	16.96	21.08	26.5	24.94	3.92	0.27
**Time Point**	**Mean Ranks of M-D Socket Dimensions**				**Post Hoc Test Statistic**	***p*-Value**
	**PRF**	**A/S-PRF**	**FDBA/CM**	**RCP**		
Day 0	19.42	24.46	23.8	20.17	1.51	0.68
Day 10	15.75	20.96	24.3	29.17	6.46	0.019
Day 21	17.04 *	17.67 *	23.55	32.67	10.16	0.017
Day 28	15.67 *	19.33	24.85	30.83	8.79	0.012

Significant changes are indicated by asterisks.

**Table 3 diagnostics-12-02403-t003:** Comparison of mean ranks of pain assessment and healing index on day 10, day 21, and day 28 among the 4 study groups.

Time Point	Mean Ranks of Pain Assessment				Post Hoc Test Statistic	*p*-Value
	A-PRF	A/S-PRF	FDBA/CM	RCP		
Day 10	14.50 *	14.50 *	31.80 *	31.11 *	26.84	<0.0001
Day 21	19.00 *	19.00 *	21.1	31.00 *	16.66	0.001
Day 28	21.5	21.5	21.5	23.89	3.78	0.286
**Time Point**	**Mean Ranks of Healing Index**				**Post Hoc Test Statistic**	***p*-Value**
	**A-PRF**	**A/S-PRF**	**FDBA/CM**	**RCP**		
Day 10	30.33 *	29.00 *	15.4	8.89 *	24.67	<0.0001
Day 21	25.42 *	31.46 *	18.1	9.17 *	20.44	<0.0001
Day 28	26.71 *	32.54 *	14.65	9.83*	25.72	<0.0001

Significant changes are indicated by asterisks.

## Data Availability

Data are available upon request from the corresponding author.

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
