# Peer review of "Soft-Tissue Healing Assessment after Extraction and Socket Preservation Using Platelet-Rich Fibrin (PRF) in Smokers: A Single-Blinded, Randomized, Controlled Clinical Trial"

_diagnostics, 2022, doi:10.3390/diagnostics12102403_

Round 1

Reviewer 1 Report

In general, this study has its merits, but the lack of a control group reduces the value of the study. Could the research be continued, and more data collected in this regard? This should be considered. However, this study brings interesting information to the clinician, especially clinical pictures are a good addition.

Introduction: Since smoking is the central outfit here, it should be described how smoking impairs healing, e.g. the effect of smoking on immune defenses. Please clarify.

Methods: In this study, only smokers have been compared with each other. A control group consisting of non-smokers adds a lot of value to the study. In your study participants were recruited according: current smoker to be defined as; 1or more cigarettes/day. However, it is known that the amount and duration of smoking can have an effect, and thus the clarification of which would also be an important matter. Please clarify.

Discussion: The discussion section is generally quite short. Furthermore, It should be highlighted, when considering previous studies, how they may have taken into account smoking and its impact.

It is good that you yourself point out the lack of a control group, but unfortunately this does not eliminate the problem itself

Author Response

Reviewer 1

Comment 1: Introduction: Since smoking is the central outfit here, it should be described how smoking impairs healing, e.g. the effect of smoking on immune defenses. Please clarify.

Response: Thank you for your valuable comment. A paragraph has been added from lines 55-59 and highlighted in yellow as requested.

References added:

          Qiu, F.; Liang, C.-L.; Liu, H.; Zeng, Y.-Q.; Hou, S.; Huang, S.; Lai, X.; Dai, Z. Impacts of Cigarette Smoking on Immune Responsiveness: Up and down or Upside Down? Oncotarget 2017, 8 (1), 268–284. https://doi.org/10.18632/oncotarget.13613.

          Skok, M. V.; Grailhe, R.; Agenes, F.; Changeux, J.-P. The Role of Nicotinic Receptors in B-Lymphocyte Development and Activation. Life Sci 2007, 80 (24–25), 2334–2336. https://doi.org/10.1016/j.lfs.2007.02.005.

          Tarbiah, N.; Todd, I.; Tighe, P. J.; Fairclough, L. C. Cigarette Smoking Differentially Affects Immunoglobulin Class Levels in Serum and Saliva: An Investigation and Review. Basic Clin Pharmacol Toxicol 2019, 125 (5), 474–483. https://doi.org/10.1111/bcpt.13278.

Comment 2: Methods: In this study, only smokers have been compared with each other. A control group consisting of non-smokers adds a lot of value to the study.

Response: Thank you for your valuable comment. We agree with you however, due to the aim of the present study being focused on the heavy smoker population, a non-smoker was not added to the present study design. Also, the PRF effect on soft tissue healing was already evaluated on non-smokers in many previous studies, which were added in the discussion part (lines 430-436) compared with findings we found on smokers.

Comment 3: In your study participants were recruited according: current smoker to be defined as; 1or more cigarettes/day. However, it is known that the amount and duration of smoking can have an effect, and thus the clarification of which would also be an important matter. Please clarify.

Response: Thank you for addressing the important point after revisiting the data. All participants smoked more than 10 cigarettes/ day, defined as heavy smokers. And smoked for more than five years.

Reference:

Agnihotri, R.; Pandurang, P.; Kamath, S. U.; Goyal, R.; Ballal, S.; Shanbhogue, A. Y.; Kamath, U.; Bhat, G. S.; Bhat, K. M. Association of Cigarette Smoking With Superoxide Dismutase Enzyme Levels in Subjects With Chronic Periodontitis. J Periodontol 2009, 80 (4), 657–662. https://doi.org/10.1902/jop.2009.080545.

Comment 5: Discussion: The discussion section is generally quite short. Furthermore, It should be highlighted, when considering previous studies, how they may have taken into account smoking and its impact.

Response: Thank you for the valuable comment. A paragraph has been added from lines 404-413 and highlighted in yellow as requested.

Reference:

Sanari, A. A.; Alsolami, B. A.; Abdel-Alim, H. M.; Al-Ghamdi, M. Y.; Meisha, D. E. Effect of Smoking on Patient-Reported Postoperative Complications Following Minor Oral Surgical Procedures. Saudi Dent J 2020, 32 (7), 357–363. https://doi.org/10.1016/j.sdentj.2019.10.004.

Kuśnierek, W.; Brzezińska, K.; Nijakowski, K.; Surdacka, A. Smoking as a Risk Factor for Dry Socket: A Systematic Review. Dent J (Basel) 2022, 10 (7), 121. https://doi.org/10.3390/dj10070121.

Comment 6: It is good that you yourself point out the lack of a control group, but unfortunately this does not eliminate the problem itself

Response: It was considered un-ethically to add a negative group by leaving an empty socket after extraction due to the compromised healing smokers usually have. Smoking has been associated with multiple post-extraction complications, mainly alveolar osteitis, and keeping the participants’ well-being in mind; the sockets cannot be left empty; thus, to minimize these complications, a collagen sponge has been chosen for a negative group it shows a hemostatic effect and aids in closing wound surfaces (10.5125/jkaoms.2015.41.1.26). Furthermore, as these sites are planned to receive implants, not augmenting them will lead to compromised results.

References:

Kuśnierek, W.; Brzezińska, K.; Nijakowski, K.; Surdacka, A. Smoking as a Risk Factor for Dry Socket: A Systematic Review. Dent J (Basel) 2022, 10 (7), 121. https://doi.org/10.3390/dj10070121.

Cho, H.; Jung, H.-D.; Kim, B.-J.; Kim, C.-H.; Jung, Y.-S. Complication Rates in Patients Using Absorbable Collagen Sponges in Third Molar Extraction Sockets: A Retrospective Study. J Korean Assoc Oral Maxillofac Surg 2015, 41 (1), 26. https://doi.org/10.5125/jkaoms.2015.41.1.26.

Reviewer 2 Report

Manuscript ID: diagnostics-1904961

Title: Comparison of Soft Tissue Healing Assessment after Extraction and Socket Preservation using Platelet‐Rich Fibrin (PRF) among Smokers. A single-blinded randomized controlled clinical trial.

1.What is the main question addressed by the research?

To evaluate the ability of platelet-rich fibrin (PRF) to enhance socket wound healing in smokers.

2.Is it relevant and interesting?

The article is relevant and interesting.

3.How original is the topic?

The topic is current.

4.What does it add to the subject area compared with other published material?

The authors have collected and analyzed a great deal of recent data.

5.Is the paper well written?

Yes, the article is well written.

6.Is the text clear and easy to read?

Yes, but minor English editing is required.

7.Are the conclusions consistent with the evidence and arguments presented?

Yes, the conclusions consistent with the evidence and arguments presented but further studies are needed to confirm these hypotheses.

8.Do they address the main question posed?

Yes, the Authors addressed the main question posed.

Other comments:

  • English language: minor English editing is required.
  • Summary of abbreviations required.
  • Introduction: This section needs some improvements. I would suggest inserting a sentence on academic debate on platelet concentrate effect [https://doi.org/10.1186/s40729-021-00393-0].
  • Materials and methods: This section has been properly prepared.
  • Results: This section has been properly prepared.
  • Discussion: A comparison with other studies analysing PRF effect in socket healing e.g. in patients under medication is missing. Please improve. The Authors may improve this section by including a reference to this study : “https://doi.org/10.1007/s00784-020-03420-3”.
  • Conclusion: This section has been properly prepared but further studies are needed to confirm Authors’ hypotheses.
  •  

After making the indicated changes, the article may be suitable for publication.

Thanks for the opportunity to review this manuscript.

Author Response

Diagnostics-1904961

Editor-in-Chief,

Prof. Dr. Andreas Kjaer

Thank you for allowing us to submit a  second revised draft of the manuscript ‘Comparison of Soft Tissue Healing Assessment after Extraction and Socket Preservation using Platelet‐Rich Fibrin (PRF) among Smokers. A single-blinded randomized controlled clinical trial.”  for publication in the journal Diagnostics. We appreciate the time and effort that you and the reviewers dedicated to providing feedback on our manuscript and are grateful for the insightful comments on and valuable improvements to our paper. We have incorporated most of the suggestions made by the reviewers. Those changes are highlighted within the manuscript. Please see below for a response to the reviewers’ comments and concerns.

Reviewer 2

Comment 1: English language: minor English editing is required.

Response:  Thank you for your valuable comment. Full English editing was done for the whole manuscript.

Comment 2: Summary of abbreviations required.

Response:  Thank you for your valuable comment. A summary of abbreviations was added at the end of the manuscript 

Comment 3: Introduction: This section needs some improvements. I would suggest inserting a sentence on academic debate on platelet concentrate effect [https://doi.org/10.1186/s40729-021-00393-0].

Response:  Thank you for your valuable comment. The suggested reference and a sentence have been added in lines 64-66, highlighted in yellow as requested

Reference:

Al-Maawi, S.; Becker, K.; Schwarz, F.; Sader, R.; Ghanaati, S. Efficacy of Platelet-Rich Fibrin in Promoting the Healing of Extraction Sockets: A Systematic Review. Int J Implant Dent 2021, 7 (1), 117. https://doi.org/10.1186/s40729-021-00393-0.

Comment 4: Discussion: A comparison with other studies analysing PRF effect in socket healing e.g. in patients under medication is missing. Please improve. The Authors may improve this section by including a reference to this study: “https://doi.org/10.1007/s00784-020-03420-3”.

Response: Thank you for your valuable comment. The suggested reference and a sentence have been added in lines 422-429, highlighted in yellow as requested

References:

Brancaccio, Y.; Antonelli, A.; Barone, S.; Bennardo, F.; Fortunato, L.; Giudice, A. Evaluation of Local Hemostatic Efficacy after Dental Extractions in Patients Taking Antiplatelet Drugs: A Randomized Clinical Trial. Clin Oral Investig 2021, 25 (3), 1159–1167. https://doi.org/10.1007/s00784-020-03420-3.

Asaka, T.; Ohga, N.; Yamazaki, Y.; Sato, J.; Satoh, C.; Kitagawa, Y. Platelet-Rich Fibrin May Reduce the Risk of Delayed Recovery in Tooth-Extracted Patients Undergoing Oral Bisphosphonate Therapy: A Trial Study. Clin Oral Investig 2017, 21 (7), 2165–2172. https://doi.org/10.1007/s00784-016-2004-z.

Round 2

Reviewer 1 Report

Thank you. Changes have been made accordingly. I do not have any further comments.